# Antiretroviral Adherence and Use of Antihypertensives, Statins, and Antidiabetics Among Elderly People with HIV: A 5-Year Real-World Study in Southern Italy

**DOI:** 10.3390/v17091212

**Published:** 2025-09-05

**Authors:** Pietro Trisolini, Simona Cammarota, Anna Citarella, Marianna Fogliasecca, Viviana Alicchio, Stefania Antonacci, Romina Giannini, Renato Lombardi, Mariantonietta Piccoli, Francesco Pomarico, Cataldo Procacci, Antonino Siniscalco, Stefania Spennato, Annalisa Saracino, Sergio Lo Caputo

**Affiliations:** 1Hospital Pharmacy Unit, IRCCS National Institute of Research “Saverio De Bellis”, 70013 Bari, Italy; pietro.trisolini@irccsdebellis.it; 2LinkHealth Health Economics, Outcomes & Epidemiology s.r.l., 80143 Naples, Italy; anna.citarella@linkhealth.it (A.C.); marianna.fogliasecca@linkhealth.it (M.F.); 3Hospital Pharmacy Unit, Vito Fazzi Hospital, Local Health Authority Lecce, 73100 Lecce, Italy; alicchio.viviana@gmail.com (V.A.); stefania.spennato@asl.lecce.it (S.S.); 4Pharmaceutical Department, Local Health Authority Bari, 70123 Bari, Italy; stefania.antonacci@asl.bari.it; 5Pharmaceutical Service, SS Annunziata Hospital, Local Health Authority Taranto, 74121 Taranto, Italy; romina.giannini@asl.taranto.it; 6Territorial Pharmaceutical Service, Local Health Authority Foggia, 71121 Foggia, Italy; lombardi.r@alice.it; 7Hospital Pharmacy Unit, A. Perrino Hospital, Local Health Authority Brindisi, 72100 Brindisi, Italy; mariant.piccoli@gmail.com; 8Hospital Pharmacy Department, Polyclinic of Bari, University of Bari, 70124 Bari, Italy; francesco.pomarico@policlinico.ba.it; 9Pharmaceutical Department, Local Health Authority BAT, 76125 Trani, Italy; aldo.procacci@aslbat.it; 10Hospital Pharmacy Unit Policlinic of Foggia, University of Foggia, 71122 Foggia, Italy; asiniscalco@ospedaliriunitifoggia.it; 11Clinic of Infectious Diseases, Department of Precision and Regenerative Medicine and Ionian Area (DiMePRe-J), University of Bari “Aldo Moro”, 70124 Bari, Italy; annalisa.saracino@uniba.it; 12Clinic of Infectious Diseases, Department of Medical and Surgical Sciences, University of Foggia, 71122 Foggia, Italy; sergio.locaputo@unifg.it

**Keywords:** HIV, antiretroviral therapy, administrative data, adherence, aging

## Abstract

Modern antiretroviral therapy (ART) has transformed HIV into a chronic, manageable condition. This retrospective analysis of administrative data from Apulia (Southern Italy) covering 2018–2023 evaluated demographic changes, ART regimen trends, adherence, and the use of antihypertensives, statins, and antidiabetics among people with HIV (PWH). Temporal trends were assessed using compound annual growth rate (CAGR). ART adherence was measured as proportion of days covered (PDC), categorized as <75%, 75–90%, and ≥90%. Over the study period, the proportion of PWH aged 18–54 declined, while those aged 55–64 and ≥65 increased (CAGRs: +10.9%, +14.3%). Use of single-tablet regimens rose from 45.1% to 79.6% (CAGR +12.1%), and integrase-based regimens increased from 52.0% to 69.0%, while protease inhibitor and multi-tablet regimens declined. Antihypertensives were the most prescribed concomitant drugs, followed by statins and antidiabetics (CAGRs: +5.8%, +9.7%, +9.5%). In 2023, 81.9% of subjects achieved PDC ≥ 90%, although lower adherence was observed in women and treatment-naïve individuals. These findings indicate a shift toward simplified, integrase-based regimens and high ART adherence, alongside a growing cardiometabolic burden. Tailored strategies are needed to support adherence, particularly in women and treatment-naïve individuals, and to address cardiovascular risk in aging PWH.

## 1. Introduction

The remarkable success of modern antiretroviral therapy (ART) has transformed HIV from a life-threatening condition to a manageable chronic disease. This therapeutic revolution has enabled most people with HIV (PWH) to achieve viral suppression and experience near-normal life expectancy [1,2]. Consequently, the demographic profile of PWH has shifted dramatically, with an increasingly older population requiring more complex care. More research is needed to understand the interplay between HIV, ageing, and comorbidities, and to develop tailored prevention and care strategies [3].

For the past two decades, the preferred regimens have been based on the combination of two nucleoside/nucleotide reverse transcriptase inhibitors (NRTIs) plus a third drug among the non-nucleoside reverse transcriptase inhibitors (NNRTIs), boosted protease inhibitors (PI/bs), and integrase strand transfer inhibitors (INSTIs) [4]. The latter are nowadays preferred because of their durable virologic efficacy, high barrier to resistance, and favourable tolerability and toxicity profiles [5]. Current international guidelines also recommend that clinicians, when choosing between regimens of similar efficacy and tolerability, use once-daily (OD) regimens for treatment-naive individuals beginning ART. They also suggest switching treatment-experienced individuals receiving complex or poorly tolerated regimens to OD regimens and using fixed-dose combinations (FDCs) and single-tablet regimens (STRs) to reduce pill burden [6].

Furthermore, modern ART options demonstrate remarkable potency and forgiveness—meaning they are more effective and resilient to occasional missed doses—compared to earlier generations of treatment [7,8]. Recent real-world analyses have challenged traditional adherence thresholds, suggesting that virologic success is achievable even when adherence falls below the historically recommended 90% level [9]. Indeed, no statistically significant difference in achieving viral suppression was found when comparing persons with adherence levels above 90% to those in the 75–90% range [9]. Despite these advancements, ensuring robust adherence remains a critical component of HIV care.

Even with effective HIV viral suppression, inflammation and immune dysregulation appear to increase risks for cardiovascular disease (CVD) in PWH compared to HIV negative population, making CVD prevention and management a critical focus in HIV care [10]. Both HIV infection itself and certain antiretroviral therapies contribute to metabolic comorbidities (hypertension, obesity, dyslipidemia, insulin resistance), underscoring the complex interplay between HIV, its treatment, and cardiovascular disease [11]. When compared to people without HIV, PWH have about a twofold higher risk of developing atherosclerotic cardiovascular disease (ASCVD), and their age at incident ASCVD diagnosis is about a decade younger [12,13]. In addition, several studies in PWH have suggested different ASCVD risk across genders, with an apparent stronger association between HIV and ASCVD in women than in men [14]. The recent landmark REPRIEVE trial—Randomized Trial to Prevent Vascular Events in HIV—highlighted the benefit of moderate-intensity statins in reducing major cardiovascular events [15,16]. Specifically, REPRIEVE found that taking pitavastatin reduces the risk of heart disease by 35% among PWH and reduces the risk of heart disease or death from any cause by 21% compared to placebo. Based on such findings, all recommendations emphasise a renewed focus on risk assessment and prevention of ASCVD in PWH through lifestyle measures, management of risk factors, and the use of statins in PWH aged 40 years and older [17].

An important challenge arising from the marked rise in age-associated comorbidities among PWH is polypharmacy. In an Italian outpatient clinic dedicated to managing this issue, PWH were prescribed an average of 4.2 additional drugs, increasing to 6.3 non-ART medications among those over 65 years of age [18]. This high prevalence of multiple concomitant medications places older PWH at increased risk of adverse drug events, drug–drug interactions (DDIs), hospitalizations, and mortality. Italian data indicate a high rate of DDIs, particularly in individuals with multidrug-resistant HIV, with 66.8% experiencing at least one DDI and 8.1% receiving contraindicated drug combination [19]. These issues highlight the need for real-world approaches to optimize medication regimens and reduce polypharmacy-related risks. This retrospective real-world study aims to evaluate demographic trends, antiretroviral therapy regimen changes, and treatment adherence among people with HIV by using administrative data from a region of Southern Italy (2018–2023). In addition, it also assesses trends in antihypertensive, statin, and antidiabetic medication use to identify opportunities for optimizing management and improving long-term clinical outcomes.

## 2. Methods

### 2.1. Data Source and Study Population

This observational descriptive study was carried out after approval from the institutional ethics committee of Apulia region. For this retrospective study, data were retrieved from the hospital and outpatient pharmacy databases of all Italian Local Health Units (LHUs) in the Apulia region (a region of approximately 4 million inhabitants in the south of Italy). To guarantee individuals’ privacy, an anonymous univocal numeric code was assigned to each health-assisted subject by the LHUs. The individual ID code allowed the electronic linkage between the databases. The anonymous code of the subject ensures the anonymity of the extracted data in full compliance with UE Data Privacy Regulation 2016/679 (“GDPR”) and Italian D.lgs. n. 196/2003, as amended by D.lgs. n. 101/2018. All the results of the analyses were produced as aggregated summaries, which could not be connected, either directly or indirectly, to any individual.

For each dispensed medication the following information was collected: anonymous individual code, sex and age, origin, diagnosis, date of dispensation, Anatomical Therapeutic Chemical (ATC) code, quantity dispensed, dose, formulation, and defined daily dose (DDD) according to the World Health Organization Collaborating Centre for Drug Statistics Methodology. In Italy, PWH have free access to all antiretroviral drugs approved by the European Medicines Agency (EMA), as part of the national healthcare system.

The study population comprised all PWH aged 18 years or older who had a diagnosis of HIV and received ART between 1 January 2018 and 31 December 2023. Individuals were excluded if they were receiving ART for indications other than HIV treatment, such as hepatitis B virus infection. For each calendar year, the date of the first ART claim was defined as the index date. Both subjects who had not been previously treated with ART, defined as having no ART use during the 12-month prior to the index date (i.e., treatment-naïve individuals, including those with prior ART experience but with a treatment interruption of ≥12 months), and those who were treatment experienced (i.e., with a pharmacy record for an ART medication during the 12-month prior to the index date) were included in the study population.

### 2.2. Study Measures

For each year of the study period (2018–2023), data on subject demographics (sex, age and ethnicity), NRTI backbone, ART regimen category and type of formulation were evaluated at the index date. Age was categorized into the following groups: 18–34, 35–44, 45–54, 55–64, and 65+ years old. The NRTI backbones were identified using ATC codes and classified into three categories: abacavir (ABC)/lamivudine (3TC), tenofovir disoproxil fumarate (TDF)/emtricitabine (FTC) and the tenofovir alafenamide fumarate (TAF)/emtricitabine (FTC). The latter was available in Italy since March 2017. ART regimen was defined as either a combination regimen of 1 core ART medication and ≥1 other ART medication within a 14-day period following the pharmacy record of the core ART medication (MTR) or 1 STR. Then, subjects were grouped into 1 of 3 categories for ART regimen: (1) INSTI-based; (2) NNRTI-based; (3) PI-based. Categories were not mutually exclusive, as the objective of the analysis was to describe prescribing trends over time; therefore, subjects receiving regimens containing more than one core class (e.g., both PI and INSTI) were counted in each category.

All individuals were followed for 1 year from the index date to assess the adherence to any ART and the use of antihypertensives, antidiabetics, and statins. Accordingly, the year 2023 was used as the follow-up period for subjects enrolled in 2022. Persons enrolled in 2023 were excluded from the analysis due to insufficient follow-up period.

Adherence to any ART was measured using proportion of days covered (PDC). The PDC was defined as the sum of non-overlapping days of supply for any ART divided by the number of days in the follow-up period (i.e., 365 days). We assumed a drug being available starting from the date of the prescription fill for the number of days calculated by dividing the total quantity dispensed by the DDD. Values for PDC range from 0 to 100% with higher values indicating higher adherence and “100%” indicating a subject who had complete ART medication adherence. In HIV, a high level of adherence to ART is required for viral suppression. Recent studies have shown that lower levels of adherence may be sufficient for newer ART regimens, as compared to 90% required for older ART regimens [9]. Therefore, the following PDC group were reported as follows: less than 75% (poor adherence), 75% to 90% and at least 90% (optimal adherence). The PDC was determined for all individuals in the study population and stratified by sex.

### 2.3. Statistical Analyses

Descriptive statistics were obtained representing categorical variables as frequencies and proportions (%) by calendar year. Chi-square test for trend was used to compare distributions of categorical variables over time. The compound annual growth rate (CAGR) was used to quantify temporal trends which can be interpreted as the average annual growth rate of the value (V) over a period of time (t years).CAGR = (V final/V begin)^(1/t) − 1

All analyses were also conducted including only treatment-naive individuals captured between 2019 and 2022. For subjects enrolled in 2019, the year 2018 was used as the baseline assessment period, whereas those enrolled in 2022 were followed up in 2023.

Statistical analyses were performed using SPSS software version 23 SPSS Inc., Chicago, IL, USA, with *p* < 0.05 indicating statistical significance.

## 3. Results

### 3.1. Subject Characteristics and ART Regimens, 2018–2023

Baseline subject characteristics at the index date, along with the most frequently prescribed ART regimens during the study period, are summarized in Table 1.

The cohort size remained relatively stable, ranging from 2781 persons in 2018 to a peak of 3102 in 2021, before decreasing slightly to 2797 by 2023. The proportion of males remained stable at approximately 73% (CAGR +0.2%; *p* = 0.963), whereas the age distribution shifted markedly. The proportions of individuals in the 18–34, 35–44 and 45–54 age groups declined with CAGRs of −9.7%, −4.2% and −7.2%). In contrast, the 55–64 and ≥65 age groups increased by +10.9% and +14.3%, respectively, leading to a rise in the mean age from 48.4 years in 2018 to 52.7 years in 2023. During this study period, STR markedly rose from 45.1% to 79.6% (+12.1% CAGR) with a marked decline in MTR (from 54.9% in 2018 to 20.4% in 2023; −18.0% CAGR). Among NRTI backbones, TAF/FTC use increased slightly (+2.4% CAGR, *p* < 0.0001), while the use of ABC/3TC and TDF/FTC dropped sharply (−25.6% and −31.4% CAGRs, both *p* < 0.0001). Regarding ART regimen categories, the use of INSTI-based regimens rose substantially (from 52.0% to 69.0%, +5.8% CAGR), whereas PI-based treatments markedly decreased (from 33.3% to 18.1%, −11.4% CAGR). Conversely, NNRTI-based regimens slightly rose from 27.5% to 28.4% (+0.7% CAGR).

Figure 1 shows the unadjusted trends in prescription rates of antihypertensive agents, statins, and antidiabetic medications among PWH over the 12-month follow-up period. Overall, antihypertensive drugs remained the most frequently prescribed concomitant medications, followed by statins and antidiabetics. The CAGRs of antihypertensive agents, statins, and antidiabetic medications prescription were 5.8%, 9.7%, and 9.5%, respectively. Male and female subgroups revealed similar upward trends (Figure 1).

Over the 2018–2023 period, the proportion of PWH with a PDC ≥ 90% achieved 81.9%, with a CAGR of +0.8%. PWH with a PDC ≥75% and <90% were 11.4% (CAGR +3.3%) of the overall population, while those with a poor adherence (PDC < 75%) declined from 10.8% to 6.7%, reflecting a CAGR of −11.3% (Figure 2). Stratified by sex, a lower proportion of women compared to men were optimally adherent (76.9 vs. 83.6%, i2022 to 2023) and a higher rate had a poor adherence (PDC < 75%: 9.8% vs. 5.6%).

### 3.2. Treatment-Naive Individuals

Overall, 708 treatment-naïve individuals (73.8% male, 42.1 mean age), captured between 2019 and 2022, were included (Table 2). The largest age groups were 18–34 years (29.9%) and 35–44 years (28.9%), followed by progressively smaller proportions in older categories: 45–54 years (23.7%), 55–64 years (14.8%), and ≥65 years (2.7%). The majority (70.5%) of persons used an STR as their index ART, 74.2% had TAF/FTC as the backbone, and 63.8% received the INSTI-based ART regimen. Antihypertensive medications were the most commonly prescribed concomitant drugs, used by 17.2% of the treatment-naïve cohort, followed by statins (6.9%) and antidiabetics (3.4%) (Table 2).

Among this population subset, 71.0% had optimal adherence (PDC ≥ 90%), 10.5% had PDC ≥ 75% and <90%, and 18.5% had poor adherence (PDC < 75%). Stratified by sex, the proportion of naïve-treatment individuals with adherence to ART ≥ 90% ranged from 60.2% among females to 74.9% among males, while poor adherence (PDC < 75%) was observed in 26.4% and 15.7–4.8% of subjects (Table 3).

## 4. Discussion

This comprehensive analysis describes trends among PWH treated with ART from 2018 to 2023, offering valuable insights into the evolution of HIV care in a real-world clinical practice in Southern Italy.

First, our findings revealed substantial changes in the age distribution of the overall HIV population over the study period, reflecting both population aging and evolving epidemiological patterns. The most pronounced shift was observed in the youngest age group (18–34 years), which declines significantly from 13.2% in 2018 to 7.9% in 2023, corresponding to a CAGR of −9.7%. This trend may reflect either a reduced incidence of HIV in younger populations or successful retention in care as PWH aged into older categories. The declines in the younger age categories were offset by substantial increases in older age groups. The proportion of individuals aged 55–64 years rose from 22.8% to 38.3% (CAGR: 9.0%), while those aged 65 and older increased from 6.0% to 11.6% (CAGR: 10.9%). By 2023, two-thirds of our cohort were aged ≥50 years or older, a cutoff commonly used to define “older” in HIV research. Our findings were in line with previous data reported by US and European reports [20,21]. In 2022, surveillance data revealed that over half of people living with HIV in the United States were aged ≥50 years and 16% of new HIV diagnoses occurred among this group [22]. In the Netherlands, Smit et al. projected that by 2030, 73% of people living with HIV will be aged 50 years or older, highlighting that HIV has become a chronic condition of aging [23]. Similarly, other Italian studies have documented an increasing age trend, with the proportion of PWH aged over 50 rising significantly over the last decade [24,25]. This demographic transition requires HIV care to move beyond simply achieving viral suppression and to prioritize management of comorbidities, careful monitoring of polypharmacy, and routine geriatric assessments, since older individuals are at increased risk for cardiovascular disease, diabetes, and other age-related condition [3,11,26]. Moreover, our cohort remained predominantly male (~73% throughout the study), consistent with the ICONA cohort data and other real-world studies on HIV [25,27,28]. The stable male-to-female ratio suggests that the observed demographic transitions are primarily driven by aging within the existing subjects rather than changes in the gender distribution.

In our study, we observed the widespread introduction of modern antiretrovirals and new treatment strategies. Previous studies reported that the use of novel agents—such as second-generation INSTIs and TAF—has led to stronger viral suppression, better tolerance, reduced resistance, fewer adverse events, and minimized drug–drug interactions [29,30]; however, these effects were not explored in the present study. We also noted a marked increase in the use of dual therapy regimens, specifically dolutegravir plus lamivudine and dolutegravir plus rilpivirine, which rose from 0.9% in the first year to 19.7% in the last year (CAGR ≈ 85%),underscoring their expanding role in current treatment strategies. In contrast, PI-based regimens demonstrated a steady decline from 33.3% in 2018 to 18.1% in 2023, with a CAGR of −11.4% (*p* < 0.0001). This decline in PI usage represents a significant shift away from what was previously considered the backbone of HIV treatment [31]. It is recognized that PI therapy is associated with serious adverse metabolic effects, including peripheral lipoatrophy, hyperlipidemia, and insulin resistance [32]. On the other hand, NNRTI-based regimens showed a modest increase from 27.5% in 2018 to 28.4% in 2023 (CAGR 0.7%, *p* = 0.016). This trend may reflect the introduction of newer NNRTI formulations with improved resistance barriers and tolerability profiles, providing additional treatment options for specific populations [33]. Furthermore, we observed that STR use increased substantially from 45.1% in 2018 to 79.6% in 2023, corresponding to a CAGR of 12.1%. These results are consistent with a recent Italian observational study (2014–2020), which also reported a rapid increase in STR use, accounting for over three-quarters of all ART prescriptions by 2020. These findings and ours confirm that clinicians are likewise favouring newer co-formulated options to reduce pill burden, improve tolerance, and enhance subject satisfaction. Notably, this consistency across the various Italian regions highlights a nationwide shift toward simplified, guideline-recommended therapies [6].

Regarding the analysis of concomitant medication, prescription rates for antihypertensives, statins, and antidiabetic agents rose steadily over the observation period, reflecting an aging population and the associated increase in age-related comorbidities. By contrast, treatment-naïve individuals showed lower rates of these concomitant medication use, likely due to their younger age and shorter duration of HIV infection, resulting in a lower cumulative metabolic impact. Statin therapy demonstrated a marked increase, particularly among women. This trend reflects increasing awareness of cardiovascular risk management in PWH, who may face elevated cardiovascular risks due to both traditional risk factors and HIV-specific factors [34]. However, several studies have shown that statins remain under-prescribed in people living with HIV [35], with under-prescribing particularly pronounced among women and individuals with HIV/HCV co-infection [36,37,38]. One explanation for statin under-utilization has been concern about drug–drug interactions between ART and statins [39]. However, newer INSTI-based regimens are associated with a lower risk of clinically significant drug–drug interactions, and the unnecessary avoidance of statin therapy may increase the risk of cardiovascular disease in people living with HIV [29]. Recent evidence from the RePRIEVE trial highlights the clinical benefit of statin therapy even in subjects with low-to-moderate traditional CVD risk profiles, reinforcing the importance of integrating lipid-lowering strategies into routine HIV care [16]. In response to the primary results of the REPRIEVE trial, which were reported in July 2023, the European AIDS Clinical Society (EACS), the British HIV Association (BHIVA), the US Department of Health and Human Services (DHHS), and the Spanish HIV Society (GeSIDA) have published guidance on statin use in people with HIV [17]. Together, these insights highlight the need for improved awareness and adherence to guidelines among healthcare providers to ensure appropriate statin selection and dosing, ultimately improving individual outcomes.

Finally, our data indicate that adherence patterns remained relatively stable throughout the study period, despite the overlap with the COVID-19 pandemic. From 2022 to 2023, approximately 81% of subjects achieved optimal adherence (PDC ≥ 90%). These findings are in line with current research indicating that, despite initial concerns, ART adherence among people living with HIV in Italy was not adversely affected by the COVID-19 pandemic, thanks to rapid adaptations in care delivery [40].

Recent real-world data from Italy show similar high adherence to ART, particularly with TAF-based regimens [27]. In addition, we found that females were lower-adherent than males (76.9 vs. 83.6, 2022 to 2023). This gender disparity has been previously reported in earlier studies on HIV adherence and may reflect sex-specific barriers to care and treatment faced by women [41,42]. Consistent with findings from other studies, treatment-naïve individuals showed a 10% lower average adherence to ART compared with the overall study population (71% vs. 81%) [43]. One possible explanation is that treatment-naïve individuals were younger than the overall study population, and several studies identified younger age as a predictor of lower ART adherence [44,45]. The findings of these studies and ours may be helpful for targeting and tailoring efforts to improve adherence to ART among persons who are newly treated with ART.

The findings from our study should be viewed in the context of study and administrative data limitations. While administrative data are extremely valuable for the efficient and effective examination of health care outcomes, they are primarily collected for business purposes rather than research purposes. The presence of a claim for a filled prescription does not indicate that the medication was consumed or taken as prescribed. Information that could influence study outcomes is not readily available in administrative claims data. This includes socioeconomic variables (such as education level and income) as well as clinical parameters specific to HIV, including viral load and CD4 count, which could serve as important indicators of disease status and surrogate markers of adherence. Moreover, since we defined treatment-naïve individuals as no ART use during the 12-month prior the index date, this group may have comprised those who initiated their first ART regimen during the study identification period and also those who had been potentially non-adherent to their prior ART medications and had a long interruption (i.e., at least 12 months) in therapy prior to restarting ART. The latter person subset may have biased our treatment-naïve population toward lower adherence. Notably, the COVID-19 pandemic, which led to a substantial reduction in new HIV diagnoses and delays in ART initiation in Italy—mainly due to reduced testing and healthcare disruptions—may have further influenced the composition of this patient group [46]. Finally, since our data come from a single Italian region, the patterns we identified may be specific to local subject characteristics and to region-specific prescribing practices.

In conclusion, our real-world analysis of PWH in Southern Italy from 2018 to 2023 revealed an aging cohort, a clear shift toward simpler, integrase-based regimens, and generally high adherence—although lower among treatment-naïve individuals and women, who may benefit from tailored support. Prescriptions of antihypertensives, statins, and antidiabetics rose alongside the burden of comorbidity. Given recent REPRIEVE trial data and updated guideline recommendations, we expect statin prescribing to increase further as clinicians align practice with evidence. However, future research is needed to assess adherence to statins, antidiabetics, and antihypertensives among people living with HIV, as the effective management of cardiometabolic comorbidities is becoming increasingly important in the context of aging and lifelong antiretroviral therapy. To achieve this, gender-specific interventions, enhanced provider education on drug–drug interactions, and strict adherence to evolving guidelines are essential for optimizing long-term cardiovascular and overall outcomes in people living with HIV.

## Figures and Tables

**Figure 1 viruses-17-01212-f001:**
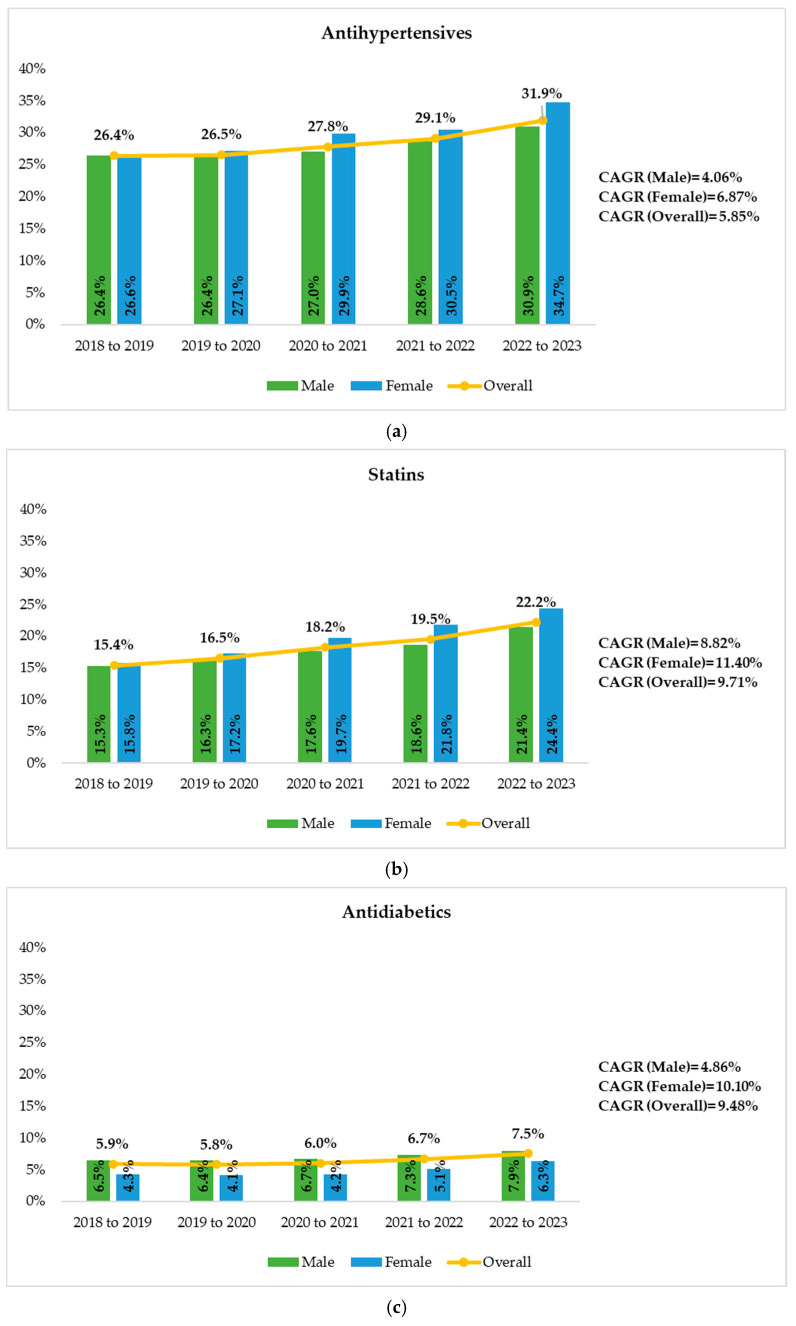
Concomitant medications for PWH by year (2018–2022). (**a**) Antihypertensives. (**b**) Statins. (**c**) Antidiabetics. Abbreviations: PWH, people with HIV; CAGR, compound annual growth rate.

**Figure 2 viruses-17-01212-f002:**
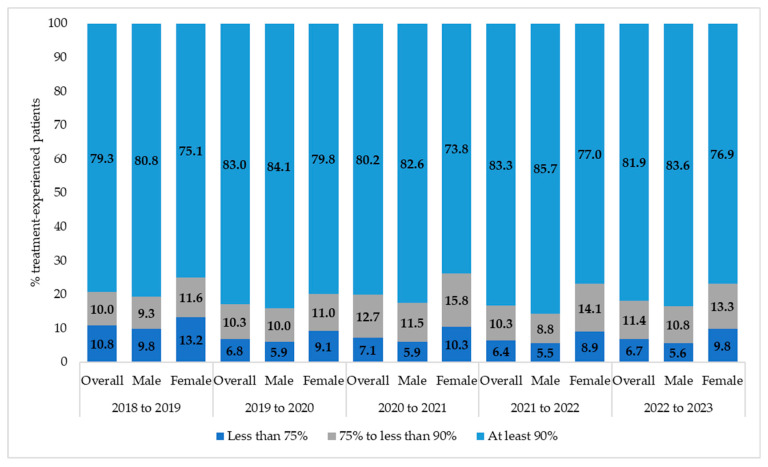
Adherence for people with HIV (2018–2022).

**Table 1 viruses-17-01212-t001:** Baseline characteristics for PWH during 2018–2023.

	2018 N = 2781		2019 N = 2957		2020 N = 3016		2021 N = 3102		2022 N = 3003		2023 N = 2797		CAGR%	*p* Value
	*n*	%	*n*	%	*n*	%	*n*	%	*n*	%	*n*	%		
Male	2029	73.0	2159	73.0	2199	72.9	2262	72.9	2212	73.7	2060	73.7	0.2	0.963
Age group														
18–34	368	13.2	400	13.5	379	12.6	367	11.8	283	9.4	222	7.9	−9.7	<0.0001
35–44	521	18.7	554	18.7	543	18.0	539	17.4	482	16.1	423	15.1	−4.2	
45–54	1092	39.3	1036	35.0	982	32.6	938	30.2	866	28.8	755	27.0	−7.2	
55–64	634	22.8	767	25.9	892	29.6	999	32.2	1078	35.9	1072	38.3	10.9	
≥65	166	6.0	200	6.8	220	7.3	259	8.3	294	9.8	325	11.6	14.3	
Non-Italian origin	124	4.5	123	4.2	123	4.1	141	4.5	110	3.7	89	3.2	−6.5	0.082
STR	1253	45.1	1551	52.5	1883	62.4	2197	70.8	2282	76.0	2227	79.6	12.1	<0.0001
MTR	1528	54.9	1406	47.5	1133	37.6	905	29.2	721	24.0	570	20.4	−18.0	
NRTI backbones														
TAF/FTC	1495	53.8	1866	63.1	2015	66.8	2054	66.2	1888	62.9	1694	60.6	2.4	<0.0001
ABC/3TC	583	21.0	558	18.9	478	15.8	416	13.4	263	8.8	134	4.8	−25.6	<0.0001
TDF/FTC	209	7.5	69	2.3	51	1.7	41	1.3	34	1.1	32	1.1	−31.4	<0.0001
None	494	17.8	464	16.7	472	17.0	591	21.3	818	29.4	937	33.7	13.6	<0.0001
ART regimen category														
INSTI-based	1445	52.0	1675	56.6	1803	59.8	1933	62.3	1995	66.4	1930	69.0	5.8	<0.0001
PI-based	925	33.3	846	28.6	780	25.9	760	24.5	655	21.8	507	18.1	−11.4	<0.0001
NNRTI-based	764	27.5	758	25.6	763	25.3	782	25.2	823	27.4	795	28.4	0.7	0.016

Abbreviations: 3TC, lamivudine; ABC, abacavir; ART, antiretroviral therapy; CAGR compound annual growth rate; FTC, emtricitabine; INSTI, integrase strand transfer inhibitor; MTR, Multiple Tablet Regimen; NRTIs, nucleos(t)ide reverse transcriptase inhibitors; NNRTIs, non-nucleoside reverse transcriptase inhibitors; PI, protease inhibitor; PWH, people with HIV; STR, Single-Tablet Regimen; TAF, tenofovir alafenamide; TDF, tenofovir disoproxil fumarate.

**Table 2 viruses-17-01212-t002:** Baseline characteristics for treatment-naïve individuals captured between 2019 and 2022 and concomitant medications during 1-year follow-up.

	Overall	
	*n* = 708	
	*n*	%
Male	523	73.8
Age group		
18–34	212	29.9
35–44	205	28.9
45–54	168	23.7
55–64	105	14.8
≥65	19	2.7
Non-Italian origin	133	18.8
STR	499	70.5
MTR	209	29.5
NRTI backbones		
TAF/FTC	526	74.2
ABC/3TC	57	8.0
TDF/FTC	22	3.1
None	104	14.7
ART regimen category		
INSTI-based	452	63.8
PI-based	171	24.2
NNRTI-based	117	16.5
Concomitant medications during 1-year follow-up		
Antihypertensives	122	17.2
Statins	49	6.9
Antidiabetics	24	3.4

Abbreviations: 3TC, lamivudine; ABC, abacavir; ART, Antiretroviral therapy; FTC, emtricitabine; INSTI, integrase strand transfer inhibitor; MTR, Multiple-Tablet Regimen; NRTIs, nucleos(t)ide reverse transcriptase inhibitors; NNRTIs, non-nucleoside reverse transcriptase inhibitors; PI, protease inhibitor; STR, Single-Tablet Regimen; TAF, tenofovir alafenamide; TDF, tenofovir disoproxil fumarate.

**Table 3 viruses-17-01212-t003:** Adherence to ART during the 12-month follow-up period in treatment-naïve individuals, overall and stratified by sex.

	Overall		Sex			
			Male		Female	
	n	%	n	%	n	%
PDC category						
<75%	131	18.5	82	15.7	49	26.4
75–90%	74	10.5	49	9.4	25	13.4
≥90%	503	71.0	391	74.9	112	60.2
Total	708	100.0	522	100.0	186	100.0

Abbreviations: ART, antiretroviral therapy; PDC, proportion of days covered.

## Data Availability

Restrictions apply to the availability of these data. The readers may contact the authors to access these data.

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
