# Peer review of "Antiretroviral Adherence and Use of Antihypertensives, Statins, and Antidiabetics Among Elderly People with HIV: A 5-Year Real-World Study in Southern Italy"

_viruses, 2025, doi:10.3390/v17091212_

Round 1
Reviewer 1 Report
Comments and Suggestions for Authors
Dear Authors, here are some comments and suggestions to improve your interesting article:
- row 82: Even with effective HIV viral suppression, inflammation and immune dysregulation appear to increase risks for cardiovascular disease (CVD) add: in PLWH compared to HIV negative population.
- Tab 1: table should contain numbers and not only the percentages. Please add the numbers in each box
- Tab 1 and Tab 2, the total of "NRTI backbones" and "ART regimens" is inferior to 100%, there is around a 15% of not classified, are they other regimens/no NRTI backbones or data not available? Please add a row to specify.
- Data show an increase in % of patients using STR over the years. Would it be possible to evaluate if there is an association between STR and high adherence?
Author Response
Dear Authors, here are some comments and suggestions to improve your interesting article:
- row 82: Even with effective HIV viral suppression, inflammation and immune dysregulation appear to increase risks for cardiovascular disease (CVD) add: in PLWH compared to HIV negative population.
We thank the reviewer for this comment. We have revised the sentence as suggested.
- Tab 1: table should contain numbers and not only the percentages. Please add the numbers in each box
We agree with the reviewer and have added the absolute numbers alongside the percentages in each cell of Table 1.
- Tab 1 and Tab 2, the total of "NRTI backbones" and "ART regimens" is inferior to 100%, there is around a 15% of not classified, are they other regimens/no NRTI backbones or data not available? Please add a row to specify.
We thank the reviewer for pointing this out. The difference is due to regimens without an NRTI backbone. We have now added a specific row in both Table 1 and Table 2 labelled “None” to clarify this point.
For the ART regimens, an outdated version of the ART-categorization variable was mistakenly reported in Tables 1 and 2. We apologize for this oversight. The tables have been updated; although some absolute counts changed, the overall trends and study conclusions remain unaffected. Moreover, the overall percentage for “ART regimens” is superior to 100% because categories (i.e. INSTI based, PI based, NNRTI based) were not mutually exclusive, consistent with our aim to describe prescribing trends over time. We have clarified this point in the Methods section (lines 160-163) as follows:
“Categories were not mutually exclusive, as the objective of the analysis was to describe prescribing trends over time; therefore, subjects receiving regimens containing more than one core class (e.g., both PI and INSTI) were counted in each category.”
Data show an increase in % of patients using STR over the years. Would it be possible to evaluate if there is an association between STR and high adherence?
We thank the reviewer for this comment. In our analysis, the increased use of STR over the study period did not appear to have a measurable impact on treatment adherence. Adherence rates remained stable over the 5-year period, with a compound annual growth rate (CAGR) of approximately 0.8%, while the proportion of patients on STR increased substantially, with a CAGR of approximately 12.1%. These findings suggest that, although STR use expanded considerably, adherence levels were already high and did not change significantly during the study period.

Reviewer 2 Report
Comments and Suggestions for Authors
Reviewer Comments to Author
I would like to thank for the opportunity to review this interesting manuscript which reports on the epidemiological trends of ART use and associated adherence as well as trends of use of antihypertensive drugs, statins and antidiabetics among PLWHIV in the region of Apulia during the period 2018-2023. This is a well-written manuscript, the methodology is sound, the results clearly presented and the findings well discussed. I suggest to address the following points would need to be addressed before publication:
-COVID-19 pandemic had a significant impact on HIV care. Implication for the study findings should be explored.
- The word patients should be changed to PLWH throughout the next if possible
-Line 65: The verb seems to be missing. e.g. “the latter are of preference”
-Line 79: Please add a reference to this statement
-Line 106: non-interventional could be omitted
-Line 129: The way treatment naïve is defined means patients who had interrupted treatment for periods longer than a year may count as naïve despite being experienced. This is discussed as a limitation but should be mentioned also in the definition.
-Line 141: How were patients receiving both PI and INSTI as core defined?
- Line 224: A dot instead of a dash
-Line 254: If possible calculating years since diagnosis could help insights into distinguishing if the pattern is due to new diagnosis or better retention to care
-Line 277: Should be rephrased as these effects were not observed in this study
-Line 285: “Increase” rather than “increasing”
Author Response
I would like to thank for the opportunity to review this interesting manuscript which reports on the epidemiological trends of ART use and associated adherence as well as trends of use of antihypertensive drugs, statins and antidiabetics among PLWHIV in the region of Apulia during the period 2018-2023. This is a well-written manuscript, the methodology is sound, the results clearly presented and the findings well discussed. I suggest to address the following points would need to be addressed before publication:
-COVID-19 pandemic had a significant impact on HIV care. Implication for the study findings should be explored.
We thank the reviewer for this insightful comment. As now specified in the Discussion section (lines 350–356), we have addressed the potential implications of the COVID-19 pandemic on our findings. Our results show that adherence patterns remained relatively stable throughout the study period, with approximately 81% of subjects achieving optimal adherence (PDC >90%) from 2022 to 2023, despite the overlap with the pandemic. These findings are consistent with previous research indicating that, thanks to rapid adaptations in care delivery, ART adherence among people living with HIV in Italy was not adversely affected by COVID-19 (Dusina et al., 2023; https://doi.org/10.1007/s10461-022-03854-8). In addition, in the Limitations paragraph (Discussion: Lines 384-386), we have acknowledged that the pandemic’s impact on HIV care—including reduced testing, fewer new HIV diagnoses, and delays in ART initiation—may have influenced the composition of our treatment-naïve cohort (Dorrucci M et al., 2023; https://doi.org/10.1093/eurpub/ckad156)
- The word patients should be changed to PLWH throughout the next if possible
We have carefully reviewed the manuscript and replaced the term patients with people living with HIV (PLWH) throughout the text.
-Line 65: The verb seems to be missing. e.g. “the latter are of preference”
Thank you for pointing this out. We have corrected the sentence by adding the verb as follows: “The latter are nowadays preferred because of their durable virologic efficacy, high barrier to resistance, and favourable tolerability and toxicity profiles.”
-Line 79: Please add a reference to this statement.
We have added the appropriate reference to support this statement, as requested.
-Line 106: non-interventional could be omitted.
We agree with the reviewer and have removed the term “non-interventional” from the sentence.
-Line 129: The way treatment naïve is defined means patients who had interrupted treatment for periods longer than a year may count as naïve despite being experienced. This is discussed as a limitation but should be mentioned also in the definition.
We thank the reviewer for this useful suggestion. We have revised the definition in the Methods section to clarify that, according to our definition, the treatment-naïve group may also include individuals who had previously received ART but experienced a treatment interruption of at least 12 months before the index date.
Methods Lines 143-148. “Both subjects who had not been previously treated with ART, defined as having no ART use during the 12-month prior to the index date (i.e. treatment-naïve individuals, including those with prior ART experience but with a treatment interruption of ≥12 months), and those who were treatment experienced (i.e. with a pharmacy record for an ART medication during the 12-month prior to the index date) were included in the study population.”
-Line 141: How were patients receiving both PI and INSTI as core defined?
We thank the reviewer for this observation. In our analysis, the variable ART regimen category was defined based on the core agent(s) dispensed at the index date. As the aim of the study was to describe prescribing trends over time, categories were not mutually exclusive. Therefore, subjects receiving regimens containing more than one core class (e.g., both PI and INSTI) were counted in each relevant category. We have clarified this point in the Methods section (lines 160-163) as follows:
“Categories were not mutually exclusive, as the objective of the analysis was to describe prescribing trends over time; therefore, subjects receiving regimens containing more than one core class (e.g., both PI and INSTI) were counted in each category.”
- Line 224: A dot instead of a dash
We thank the reviewer for noticing this. We have replaced the dash with a dot as suggested.
-Line 254: If possible calculating years since diagnosis could help insights into distinguishing if the pattern is due to new diagnosis or better retention to care.
We thank the reviewer for this valuable suggestion. Unfortunately, the date of HIV diagnosis was not available in our dataset; therefore, we were unable to calculate years since diagnosis and explore this aspect further.
-Line 277: Should be rephrased as these effects were not observed in this study.
We thank the reviewer for this observation. We have rephrased the sentence to clarify that the reported effects are derived from previous literature as follows:
Discussion lines 300-303 “Previous studies reported that the use of novel agents—such as second-generation INSTIs and TAF—has led to stronger viral suppression, better tolerance, reduced resistance, fewer adverse events, and minimized drug–drug interactions; however, these effects were not explored in the present study-”
-Line 285: “Increase” rather than “increasing”
We thank the reviewer for the suggestion. We have replaced “increasing” with “increase” as suggested.

Reviewer 3 Report
Comments and Suggestions for Authors
Dear authors,
congratulations on your valuable work. Please, find here below some suggestions to further improve the quality of your paper.
- You can remove the study period from the title and just consider to mention it in the methods. You may cite the 5-year study period in the title, but I would remove the time period.
- Since your work focused on a real-life setting, I would recommend to add some new evidence from similar settings when presenting your context in the introduction. In particular, I recommend to add a specific paragraph about the polypharmacy issues in PLWHIV and real-world evidence related to the optimization of the therapy. See, as examples, the following references: https://doi.org/10.3389/fphar.2025.1633968; doi:10.1093/jac/dkae190.
- Beware of the term "patient" in this context. Please, revise the whole manuscript and change it to more neutral terms like subject, individual, person,...
- Since you considered combined back-bones only, you decided not to include the so-called dual therapies in your analysis. I would recommend to also take them into consideration in your study. If it is not possible due to limitations, please add a specific paragraph in the discussion.
Author Response
- You can remove the study period from the title and just consider to mention it in the methods. You may cite the 5-year study period in the title, but I would remove the time period.
We thank the reviewer for the suggestion. As advised, we have removed the specific study period from the title and instead referred to the study duration more generally. The title has been revised to: "Antiretroviral adherence and use of antihypertensives, statins, and antidiabetics among elderly people with HIV: a 5-year real-world study in Southern Italy."
- Since your work focused on a real-life setting, I would recommend to add some new evidence from similar settings when presenting your context in the introduction. In particular, I recommend to add a specific paragraph about the polypharmacy issues in PLWHIV and real-world evidence related to the optimization of the therapy. See, as examples, the following references: https://doi.org/10.3389/fphar.2025.1633968; doi:10.1093/jac/dkae190.
In line with the comment, we have enriched the Introduction by adding a specific paragraph addressing polypharmacy issues in PWH and providing real-world evidence related to therapy optimization, with specific reference to the Italian context. The new paragraph (lines 103–112) reads as follows:
“An important challenge arising from the marked rise in age-associated comorbidities among PWH is polypharmacy. In an Italian outpatient clinic dedicated to managing this issue, PWH were prescribed an average of 4.2 additional drugs, increasing to 6.3 non-ART medications among those over 65 years of age [Cattaneo, D., Oreni, L., Meraviglia, P. et al. Polypharmacy and Aging in People Living with HIV: 6 Years of Experience in a Multidisciplinary Outpatient Clinic. Drugs Aging 40, 665–674 (2023). https://doi.org/10.1007/s40266-023-01037-1]. This high prevalence of multiple concomitant medications places older PWH at increased risk of adverse drug events, drug–drug interactions (DDIs), hospitalizations, and mortality. Italian data indicate a high rate of DDIs, particularly in individuals with multidrug-resistant HIV, with 66.8% experiencing at least one DDI and 8.1% receiving contraindicated drug combination [Mazzitelli, M., Pontillo, D., Clemente, T., Di Biagio, A., Cenderello, G., Rusconi, S., Menzaghi, B., Fornabaio, C., Garlassi, E., Zazzi, M., Castagna, A., & Cattelan, A. (2024). Polypharmacy, anticholinergic burden and drug-drug interaction assessment in people with four-class-resistant HIV: data from the PRESTIGIO registry. The Journal of antimicrobial chemotherapy. https://doi.org/10.1093/jac/dkae190.]. These issues highlight the need for real-world approaches to optimize medication regimens and reduce polypharmacy-related risks.”
- Beware of the term "patient" in this context. Please, revise the whole manuscript and change it to more neutral terms like subject, individual, person.
We thank the reviewer for the observation. As suggested, we have carefully revised the entire manuscript and replaced the term “patient” with more neutral alternatives such as “person,” “individual,” or “subject” according to the context.
- Since you considered combined back-bones only, you decided not to include the so-called dual therapies in your analysis. I would recommend to also take them into consideration in your study. If it is not possible due to limitations, please add a specific paragraph in the discussion.
We thank the reviewer for this valuable suggestion. Although the Results section reports ART regimen category at a class level (e.g., INSTI-based) without detailing individual regimen types, we explored dual therapy regimens for descriptive purposes. Their prevalence increased from 0.9% in the first year to 19.7% in the last year, corresponding to a compound annual growth rate (CAGR) of approximately 85%. This information has been added to the Discussion section as “data not shown” to highlight the growing role of dual therapies in clinical practice.
Discussion Lines 303-307. “We also noted a marked increase in the use of dual therapy regimens, specifically dolutegravir plus lamivudine and dolutegravir plus rilpivirine, which rose from 0.9% in the first year to 19.7% in the last year (CAGR ≈ 85%) (data not shown), underscoring their expanding role in current treatment strategies.”
